# Research priorities of caregivers and individuals with dementia with Lewy bodies: An interview study

Melissa J. Armstrong[1]*, Noheli Gamez[1], Slande Alliance[1], Tabassum Majid[2], Angela Taylor[3], Andrea M. Kurasz[4], Bhavana Patel[1], Glenn Smith[4]

**1** Department of Neurology, University of Florida College of Medicine, Gainesville, Florida, United States of America, **2** Erickson School of Aging Studies, University of Maryland, Baltimore County, Baltimore, Maryland, United States of America, **3** Lewy Body Dementia Association, Lilburn, Georgia, United States of America, **4** Department of Clinical and Health Psychology, University of Florida, Gainesville, Florida, United States of America

* melissa.armstrong@neurology.ufl.edu

**Data Availability Statement:** All relevant data (e.g. coding tree, exemplar quotes) are within the manuscript and its Supporting Information files.

## Abstract

### Background

Funding bodies are placing increased emphasis on patient and public involvement in research, but the research priorities of individuals and caregivers living with dementia with Lewy bodies (DLB) are unknown.

### Method

Investigators conducted telephone interviews with individuals living with DLB and caregivers. Participants were recruited from a Lewy Body Dementia Association Research Center of Excellence. Interviews employed a semi-structured questionnaire querying research needs in different categories and then asking participants to select their top priorities. Investigators used a qualitative descriptive approach to analyze transcripts and identify themes.

### Results

Twenty individuals with DLB and 25 caregivers participated. Seventeen from each group participated as part of a patient-caregiver dyad. Twenty-three of the caregivers were spouses, two were daughters. Individuals with DLB and caregivers identified research needs relating to focusing on awareness, determining the cause of DLB, improving diagnosis, and investigating what to expect/disease stages. Participants also highlighted DLB symptoms needing additional research, therapies to prevent, cure, or slow the progression of DLB, and research targeting daily function and quality of life, caregiving, and improving education.

### Conclusions

These findings support the research priorities defined in the National Institutes of Health dementia care summits in addition to ADRD priority-setting summits. Research is needed

**Funding:** This work was supported in part by the Agency for Healthcare Research and Quality (AHRQ K08HS24159, PI: MJA) and the National Institute on Aging (NIA AG047266, PI: T. Golde). Lewy body dementia research at the University of Florida is supported by the University of Florida Dorothy Mangurian Headquarters for Lewy Body Dementia and the Raymond E. Kassar Research Fund for Lewy Body Dementia. The funders had no role in study design, data collection and analysis, decision to publish, or preparation of the manuscript.

**Competing interests:** MJA receives research support from ARHQ (K08HS24159), a 1Florida ADRC pilot grant (AG047266), the Florida Department of Health Ed & Ethel Moore research program, and as the local PI of a Lewy Body Dementia Association Research Center of Excellence. AT is employed by the Lewy Body Dementia Association. BP receives research support from an American Academy of Neurology Clinical Research Training Scholarship in Lewy Body Dementia and received compensation for consultation with Medtronic. GS receives research support from the 1Florida ADRC (AG047266) and the Florida Department of Health Ed & Ethel Moore research program. NG, SA, TM, and AMK have no competing interests to report. This does not alter our adherence to PLOS ONE policies on sharing data and materials.

across all domains of DLB. Funding should be informed by the priorities of all relevant stakeholders and support research investigating causes, natural history, biomarkers, and treatment in addition to research targeting themes regarding living with disease (e.g. independence, quality of life, caregiving, and education).

## Introduction

Funding bodies including the National Institutes of Health (NIH) are placing increased emphasis on patient and public involvement in research. Patients can have a role throughout the research process, from study design to dissemination [1, 2]. Before studies are even initiated, patients, caregivers, and the public have a role in identifying research priorities [2]. This is critical given a mismatch between the research priorities of patients, caregivers, clinicians, and researchers [3].

The Advisory Council on Alzheimer's Research, Care and Services recommends that all funders "should establish the engagement of the AD/ADRD [Alzheimer disease/ Alzheimer-disease related dementia] community as a standard practice in. . . participating in setting national research priorities for AD/ADRD. . ." [4]. The NIH has adopted multiple strategies to increase the role of patient and public involvement in dementia priority setting. ADRD national research priority summits include sessions led by nongovernmental organizations to increase non-expert stakeholder input and public-private partnerships [5]. The 2017 National Research Summit on Care, Services, and Supports for Persons with Dementia and Their Caregivers included "persons living with dementia" and family caregiver stakeholder groups as two of the six stakeholder groups informing summit planning and recommendations [6]. One aim of the planned 2020 National Research Summit on Care, Services, and Supports for Persons with Dementia and their Caregivers was to develop recommendations for research priorities to inform federal agencies and other research funders, but the in-person meeting was cancelled due to the global COVID-19 pandemic. The summit will now occur via a series of virtual meetings in summer 2020.

Involvement of patient representatives alongside experts in priority setting reflects a "participation" engagement strategy. Through participation, representatives have an active and equal voice in the process. Participation strategies usually rely on small numbers of representatives, potentially missing other perspectives. Consultation strategies (e.g. surveys, interviews, focus groups) collect a variety of views from larger groups of people, but fail to give patient representatives an active voice in the process. Employing participation and consultation approaches as complementary may be the optimal approach to incorporate the views of a diverse group while also giving patient representatives an equal and active voice in decision making [7].

Stakeholders representing individuals with Lewy body dementia participated in NIH AD/ADRD priority setting [5] and the 2017 care summit [6]. However, no prior research investigates the research priorities of individuals and caregivers living with dementia with Lewy bodies (DLB). DLB is a subset of Lewy body dementia, the 2nd most common neurodegenerative dementia in the United States [8]. DLB is a dementia with clinical and pathological overlap with Parkinson disease (PD) and both fall in the pathological category of Lewy body diseases. However, DLB has important differences from both PD and AD/other dementias which could meaningfully impact research priorities. For example, symptoms such as cognitive fluctuations, hallucinations, and dream enactment behavior distinguish individuals with DLB from

AD [9]. Individuals with clinically-diagnosed DLB survive a median of 3–4 years after presentation [10–12], shorter than individuals with PD [13, 14] or AD dementia [11, 15]. Caregiver burden in DLB may be similar to PD dementia [16], but it is higher in DLB compared to AD dementia [17–19]. Quality of life is also worse in DLB compared to AD dementia [20, 21]. Given these and other important differences between DLB and other dementias and parkinsonisms, prior publications reporting on research priorities in PD [22, 23] and dementia in general [24–27] may not capture the priorities of individuals with DLB. We thus aimed to identify the research priorities of individuals with DLB and caregivers, particularly to guide research at our center.

## Methods

### Study design

The study used telephone interviews to investigate the research priorities of individuals living with DLB and caregivers of individuals living with DLB. Investigators used a qualitative descriptive approach [28] to analyze interview transcripts. A qualitative descriptive approach involves reporting and summarizing straightforward accounts of participants' views without an intent to generate or test theory. A qualitative descriptive approach was employed given the aim of identifying research priorities, rather than investigating theories underpinning those priorities. Consolidated criteria for reporting qualitative research guided study reporting (S1 Checklist) [29].

### Population and recruitment

Individuals with DLB and caregivers of individuals with DLB were recruited from the UF Health Norman Fixel Institute for Neurological Diseases, a Lewy Body Dementia Association Research Center of Excellence. Inclusion criteria were: (1) patient or caregiver of a patient followed at the Fixel Institute, (2) personal diagnosis of DLB [9] or caregiver for an individual diagnosed with DLB, (3) if a person with DLB, clinician-judged mild-moderate severity such that the individual could understand the study and provide personal opinions, and (4) willingness to participate in a telephone interview. Individuals and caregivers were not required to participate as a dyad. Participants were recruited when presenting for routine clinical visits or through a consent-to-contact research database at the institute. The University of Florida institutional review board provided approval (IRB201500996). The study used a waiver of documentation of informed consent.

### Data collection and analysis

The PI (MJA), a DLB specialist, drafted the semi-structured interview guide (S1 File) and revised it based on suggestions from a neuropsychologist (GS), a dementia specialist (not further involved in the project), PhD specializing in qualitative research (TM), and former caregiver of an individual with DLB (AT). The interview included questions regarding clinical care priorities, which will be analyzed separately. Research questions asked about participant priorities for DLB research in general and then specifically queried participant priorities regarding research on DLB symptoms, daily challenges, caregiving/family life, and diagnosis. Finally, the interview guide asked the participant to identify how they would divide $1000 to spend on DLB research and to prioritize what research was most important to them.

A research assistant with qualitative research experience conducted the telephone interviews (SA). The research assistant had no relationship with participants prior to the study. Interviews occurred via telephone to allow individuals to participate from home. Participants

selected the interview times. The preferred approach was to interview individuals with DLB and their caregivers separately (if both participated), but the interviewer accommodated requests for the caregiver to be present during the patient interview. Any caregiver opinions offered during a patient interview were coded as belonging to the caregiver, not the individual with DLB. Audio recording started after allowing for participant questions and verbal consent at the beginning of the call. A professional service transcribed the interviews verbatim, so member checking was not performed.

Investigators used tables in Microsoft Word® and Excel 2016® to organize data and a qualitative descriptive approach to identify and organize themes [28]. Broad topics/categories were defined by interview questions, but themes were identified from interview transcripts. The PI and two research assistants independently analyzed interview transcripts to create a codebook and then reached consensus regarding emerging themes (open coding). The research assistants analyzed remaining transcripts using a constant comparative technique, revising themes and subthemes with the PI if needed (axial coding) (S2 File). Coders assessed saturation during analysis. Co-investigators gave feedback after the initial coding. Participants were numbered in the analysis such that participants who enrolled in the study as a dyad shared participant numbers (with "P" indicating patient participants and "CG" indicating caregiver participants).

## Results

### Demographic and interview characteristics

Interviews occurred between 1/22/2018 and 5/6/2019. Twenty individuals with DLB and 25 caregivers participated (Table 1). Seventeen from each group participated as part of a patient-

**Table 1. Participant demographics.**

| | Individuals with dementia with Lewy bodies (n = 20) | Caregivers of individuals with dementia with Lewy bodies (n = 25) |
|---|---|---|
| Gender (number, % male) | 18 (90%) | 3 (12%) |
| Age range (number, %) | | |
| 50–59 | 1 (5%) | 8 (32%) |
| 60–69 | 10 (50%) | 9 (36%) |
| 70–79 | 7 (35%) | 6 (24%) |
| >80 | 2 (10%) | 2 (8%) |
| Race/Ethnicity | | |
| White non-Hispanic | 17 (85%) | 23 (92%) |
| White Hispanic | 1 (5%) | 0 (0%) |
| Black | 2 (10%) | 2 (8%) |
| Highest education | | |
| Did not finish high school | 1 (5%) | 1 (4%) |
| High school graduate (or equivalent) | 1 (5%) | 1 (4%) |
| Associate degree or vocational training | 2 (10%) | 5 (20%) |
| Some college, no degree | 3 (15%) | 4 (16%) |
| Bachelor degree | 5 (25%) | 12 (48%) |
| Advanced degree (e.g. MS, MD, PhD) | 8 (40%) | 2 (8%) |
| Relationship of caregiver to individual with dementia with Lewy bodies | | |
| Spouse | | 23 (92%) |
| Child | | 2 (8%) |

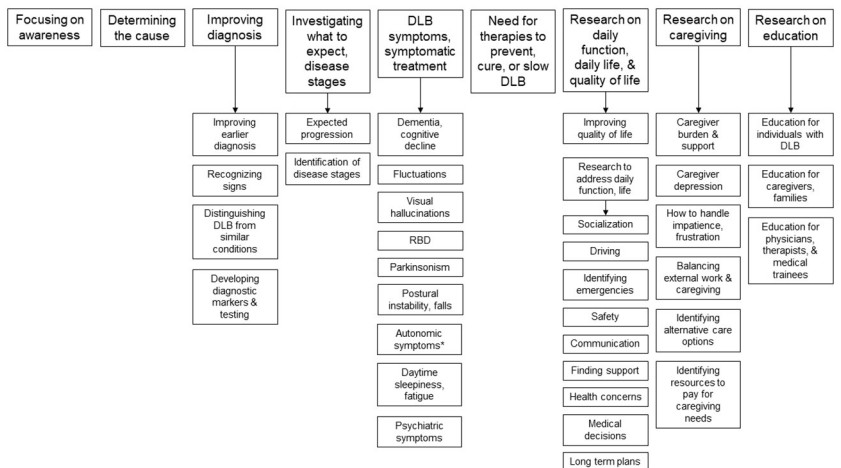

**Fig 1. Areas needing research as identified by individuals with dementia with Lewy bodies and caregivers.** Key research-related themes identified in interviews with individuals with dementia with Lewy bodies and caregivers are presented in the top row of boxes. Key subthemes relating to identified themes are presented underneath the relevant research themes.

caregiver dyad. Of the 23 participating spouses, 20 were wives. Both participating children were daughters.

Average interview duration was 28:11 minutes for individuals with DLB and 37:10 minutes for caregivers. Caregivers helped individuals with DLB participate in two interviews. Participants confirmed that all aspects of DLB need research. Specific categories of research included focusing on awareness, determining the cause, improving diagnosis, investigating what to expect (i.e. natural history) and disease stages, DLB symptoms needing additional research, the need for therapies to prevent, cure, or slow DLB, daily function, daily life, and quality of life, caregiving, and improving education (Fig 1). Exemplar quotes supporting these themes are provided below.

## Focusing on awareness

Multiple caregivers described a desire to see research focus more on increasing awareness of DLB:

> *So research I think could focus more on awareness. . . give it the same attention or least some of the attention that they do other areas that seem to overshadow this. Cuz I'd never heard of it. When I went in there and they said he had Lewy body, and I'm like, "What is that?" "What does that mean?" You know? It was like some rare disease I'd never heard of, and I know now a lot of people have it.*

(CG8, wife)

## Determining the cause

One individual with DLB and approximately half of caregivers discussed the need for more research on the cause of Lewy body disease:

> *Maybe this is where we should be starting first, maybe step back a little bit and say, you know, "Let's look at maybe this'd be the cause of the—if a light went out, what made the light go out?*

*Well, maybe the breaker tripped. Maybe the bulb burned out. Maybe the wire broke." You know, there's a lotta maybes.*

(P4)

*Well, you've got to find the root cause. . . You know, you find the cause or the problems, and you can treat it or correct it.*

(CG21, daughter)

In addition to desiring to know causes in general, multiple participants wanted further research regarding whether other health problems, prior injuries (e.g. head trauma), exposures (e.g. from mining), a history of shift work, or a traumatic event could contribute to developing DLB.

## Improving diagnosis

Multiple individuals with DLB and almost half of caregivers discussed the importance of research to improve early diagnosis (Table 2). While these respondents discussed the need for earlier diagnosis, particularly in circumstances where a diagnosis was delayed, one individual

**Table 2. Research priorities relating to diagnosis.**

| Theme | Exemplar Quotes |
|---|---|
| Improving earlier diagnosis | Of course, and the earlier—the earlier it could be, noticed, the symptoms noticed and diagnosed, I think it would be the—certainly to the patient's advantage, as well as the researcher, the physician involved. (P16) |
| | If it could be diagnosed earlier, yes, it would—I mean, it would help. (CG23, wife) |
| Recognizing signs | Maybe there oughta be a way research can—I don't know if they can do it chemically or what, but, if it could get the telltale signs so even at your regular physician that you go to day to day to day could pick up on it. (P4) |
| | To determine what causes it and how to recognize those signs. I mean, I have read a lot now. I didn't initially recognize the signs. (CG2, daughter) |
| Distinguishing DLB from similar conditions | More research into the diagnosis and being able to state that that's what somebody truly has. I know there's no diagnostic test and all of that upfront, but better definitions so people leave with saying, "Yeah, that's what it is. I can throw myself into this now." (CG13, wife) |
| | How it's associated with Parkinson's and—because there's so many things that are similar that started out as a Lewy body and then when you get a Parkinson's diagnosis, you're thinking, "Okay, well it was Parkinson's. He has no dementia." Then, when it's flipped around, and it started with dementia, with parkinsonism, I think trying to get that more research in that to try to grasp that and what the stages and how they're similar. (CG12, wife) |
| | More about the disease itself—and diseases related to it. How it's different from. . . different as a separate entity of dementia. (CG14, wife) |
| Developing diagnostic markers and testing | So from the start, I think it would be very nice if somebody who's at the early stages of presenting symptoms like I was a few years ago, could be subjected to some kind of imaging, technology that would be able to diagnose that immediately and determine, you know, how advanced the disease is. (P28) |
| | I wish there's some way that you could take something from Lewy body people, in, I don't know, blood, or whatever, and see what the heck's goin' on in their brain. I know that you really can't get a diagnosis unless—until you have an autopsy done. Well, if that's true, then I think it should be researched somehow that maybe x-rays or MRIs or, you know, what—PET scans, you know, I don't know, could identify it in a better—you know, how are we sure that's really what he has? (CG4, wife) |

with DLB and two caregivers didn't see value in earlier diagnosis without disease-modifying treatments:

> *Without some kind of remediation, I don't think it [early diagnosis] is valuable.*
>
> (P28)

> *Until you can actually do something about it, that just lengthens the time that you have this thread hangin' over your head. So earlier diagnosis, I'm not sure matters as long as. . . there's no cure.*
>
> (CG22, husband)

Multiple individuals with DLB and caregivers described needing more research to help individuals with DLB, caregivers, and healthcare professionals recognize the signs of DLB (Table 2). Caregivers wanted research to focus on improving the accuracy of diagnosis, including distinguishing DLB from PD and dementia generally, particularly given widespread confusion regarding Lewy body terminology. Several individuals with DLB and caregivers desired more research into tests and biomarkers to assist with DLB diagnosis (Table 2). A small minority of individuals with DLB and caregivers desired screening tests to identify at-risk individuals.

## Investigating what to expect and disease stages

A common theme amongst both individuals with DLB and caregivers was the need for research into what to expect with DLB progression:

> *l- looks at- would predict accelerating dementia so that it gives me a little bit of time in my pocket, if I'm progressing, to know that, and to know what to anticipate. So the big statistical analysis might be the right thing. . . this is what we expect his or her course will be.*
>
> (P18)

> *I guess more on the stages, an' I don't know that they really know for sure, stages of it, how it progresses, an' progression time.*
>
> (CG7, wife)

In addition to desiring information on what to expect with disease progression in general, multiple participants described wanting research to identify stages of DLB.

> *I see that literature about the seven stages of dementia with Lewy body, and I think knowing more about the later stages now would be helpful.*
>
> (P20)

> *Researching other people and see what stages—you know, seeing what stages their loved one is in and comp-, I mean, not compare but see, you know, the number of years. . . They say every person is different, but I guess if you gathered research and looked at it, you would see and kinda know what to expect, you know, at certain stages.*
>
> (CG15, wife)

## DLB symptoms needing additional research

Participants highlighted the need for more research for many of the core and supportive features of DLB (Table 3) [9].

Many discussed that research should investigate the symptoms and also medications to help:

*Medicine- trying to find the right combination.*

(P7)

*Anything you can give him to tamp down his symptoms—anything you can give to make it so it's easier for him to exercise, it's easier for him to live his daily life, is very important.*

(CG19, wife)

Participants also desired research on non-pharmacologic strategies to address DLB symptoms, including therapy, exercise, meaningful activities, "natural" therapies, changes in diet and nutrition, and unconventional approaches such as hyperbaric chambers.

*Look at people who have, a consistent habit of, aerobic exercise—versus people who don't. Yeah, see if it makes a difference.*

(CG17, wife)

*Nutrition and vitamin supplements, and chemical. . . And mostly because it has not been addressed.*

(CG6, wife)

Caregivers often wanted research on how to practically handle DLB symptoms in day-to-day life:

*I'm very interested in how the brain works and what happened and how it all progresses. . . but I have to deal first with the practical.*

(CG17, wife)

*Guidelines for when you should do different things, you know. Like, when is it appropriate to take over the decision-making?*

(CG22, husband)

*How can I tell when one of these [fluctuation] episodes is coming on?*

(CG5, husband)

*Sometimes I'm left feeling very lost on how to personally, emotionally, physically hands on deal with the hallucinations.*

(CG8, wife)

**Table 3. Research needs relating to DLB symptoms.**

| Core Feature | Exemplar Quotes |
|---|---|
| Dementia/cognitive decline | Short term memory loss. That's the big one. (P9) |
| | His memory. His mind. It's what bothers him the most, I think. He owned his own business, and he was a real whiz at figures. . . And he can hardly add anything now together. His addition is bad. His subtraction is bad. Just his, it's his mind. The dementia part—is what I really notice the most. (CG16, wife) |
| Fluctuations | I don't understand why some days I'm just so bad, and I don't remember people and stuff, and then other days I'm just—I seem alright to me. I seem just fine and other people said, "Man I wouldn't even know you have that disease." (P8) |
| | But his primary symptom that's very distressing to him is. . .cycles through some kind of shutdown and then he's wide awake and then—it's not sleeping. He's not asleep. He's awake. His eyes are wide open, but he cycles through this and it's distressing to him. He's gasping for air and he is totally helpless during this time period. That is the most distressing thing that has been going on for the past five years. Independent of dose, frequency, dosage form, additional medication, all of this, some inherent cycling through something that are causing physical symptoms. . . (CG13, wife) |
| Visual hallucinations | Well, for one thing, the people at night, hallucinations. . . if anything was to get me down at this point, it would be that. It's just—I can believe it because everybody tells me, you know, that's the way it is. But when I— when I see it, it overrides what the people tells me. (P11) |
| | Well, right now, of course, it's the hallucinations and the way he looks and sees things. The hallucinations are terrible for him. (CG15, wife) |
| REM sleep behavior disorder | The REM sleep disorder. Because that's one of the things that they have to have Lewy Bodies, and Lewy Bodies, from what I understand, have only been discovered in the past 20 years. It's been named and figured out what in the world it is. I think a lot more research needs to go in. . . if they could have more research on that REM deficiency, they may be able to link it better, going forward. (CG12, wife) |
| | Whether more people have the same issue I have with the sleep. I don't know if that's been addressed because they seem to—the only drug they seem to be able to give you are anti-psychotic or something of that nature. And that's the ones that zombie me out. (P8) |
| Parkinsonism | I know one of the things that occurs with me is the shaking—the tremors. That area that, research, I think I would encourage. (P3) |
| | And sometimes my wife says, she said this morning, "You're shuffling around." And I said, "No I'm not." But I obviously was. And I don't know if they can—I don't know how many studies they've done on that. So I can't really say, but—it seems an issue. (P8) |
| | The moment that you show—the physical alteration show up, there is a really big part of the disease that you have missed studying. (CG18, wife) |
| | He's in this slow-mo. . . You know, everything's a little slower. You know, tha movement issue, I guess—that comes with this also, that slows down your ability to just balance, and walk. (CG24, wife) |
| **Supportive Clinical Features** | |
| Postural instability, repeated falls | One of the biggest challenges I have we've kind of moved beyond falls a little bit. Now, we're just more into mobility when another person becomes responsible for helping you be mobile. So, I'm not sure if that comes under a research category, but it would definitely be something that would be helpful to know more about. (CG13, wife) |

(*Continued*)

**Table 3.** (Continued)

| Core Feature | Exemplar Quotes |
|---|---|
| Orthostatic hypotension | Blood pressure is low. Blood pressure. . . Of course, I'm starting to feel a little bit dizzy from time to time. I have to really think about getting up slower, and I don't walk down the road anymore, they was worried about me walking down the road. (P25) |
| | Well, I guess we could use the autonomic nervous effects there, I guess. Well, it's like, the blood pressure dropping. You know, which can cause falls which are the worst. (CG9) |
| Urinary incontinence | Well, of course, the incontinence. You know? And that's not something that he feels good about. He understands what's going on with that and he doesn't like it, but he doesn't have a choice, you know, but to let me help him. But I have to fight for him to let me help him for a good while, you know? (CG3, wife) |
| Constipation | She does have issues with constipation, an' you know, most o' the drugs and things we give her do help with that, but again, it's another drug that has side effects. . . (CG2) |
| Daytime sleepiness and fatigue | Daytime sleeping or need to nap. Well, when you don't sleep more than four or five hours a night, you get sleepy in the daytime. (P25) |
| | I think for him, the worst thing is that he's always tired. If you asked him what's the worst thing, he would say, "I'm always tired." (P19, wife) |
| Psychiatric symptoms (depression, anxiety, behavior changes) | It would be nice to have a magic pill that makes me- my very bad moods go away. (P28) |
| | He suffers a lot from depression and anxiety also. . . If they could address the depression and the anxiety, some of the other symptoms might not be as severe. (CG23, wife) |
| | I think the one that affects him the most is the outbursts of anger. Like, he can just be fine, and then he'll just set off into this rage. And he'll eventually get over it, but sometimes it's hours before he gets over it. . .. Well, that part of it would be the most important to me, too, because he's not that kind of a person. If they could just somehow get control over that. But again, I—you know, I don't even know what kind of research could do that. (CG4, wife) |

## Need for therapies to prevent, cure, or slow DLB

Multiple individuals with DLB and caregivers discussed the need for pharmacologic or non-pharmacologic disease-modifying therapies that will prevent, cure, or slow DLB:

*How it could possibly be prevented or mitigated or slowed down in its decline.*

(P4)

*If we know what causes it, there could be preventative care, kinda like, you know, heart trouble. We know we gotta have a low-calorie, low-cholesterol diet. We know we need to exercise.*

(CG2, daughter)

*Is there some interventional drug therapy or other therapies, boxing and, you know, whatever—that we could be doing? And if so, I think most importantly, what does that really buy us? If we go down this path, what are we really getting? Is it an extra week? Is it an extra year? And is that year really going to be better, or is it just prolonged?*

(CG5, husband)

*It's something that I wish they had a magic pill for to fix. You know, to get somebody back to where they were, cuz he's only 63.*

(CG7, wife)

## Research on daily function, daily life, and quality of life

Multiple participants emphasized that research needs to focus on daily function:

*Probably, people could live without knowing about how much your brain has shrunk if they're told you have this degenerative disease. . . I would like to know more about it like how much has your brain shrunk that causes this, but the biggest issue is how are we gonna get through the day? What's gonna be our big challenge today?*

(CG13, wife)

*I think it would be important—with research on—on your daily functions because that's important to all of us. It's like our daily functions. . . [My husband's] business required driving. And we had a small farm, and that required a lot of walking and feeding cattle. And he can't drive anymore, and he got to where he could not feed the cattle, and I'd say your daily functions are very important.*

(CG16, wife)

Individuals with DLB and caregivers offered many topics pertaining to daily life that they felt were deserving of more research, including how to maintain socialization, maintain the ability to drive, identify emergencies, keep individuals with DLB safe, communicate well between couples (patient-caregiver), find support (e.g. good healthcare providers, external programs), handle non-DLB healthcare issues, make medical decisions, and strategize long-term planning. Research aiming to improve quality of life was another common theme, particularly from caregivers:

*I wish mine had his quality of life back. I really do because he doesn't have that anymore. He had to quit work. He's on disability. He can't drive.*

(CG7, wife)

*Quality of life, that's the boat I really wanna jump on. But it might not happen in our lifetime. But maybe what you're doin' today will help somebody else.*

(CG25, wife)

## Research on caregiving

Several individuals with DLB and most caregivers identified the need for additional research addressing different realms of caregiving, including caregiver burden and support, depression and how to handle impatience and frustration, balance external work and caregiving, identify alternative care options and ways to keep the individual with DLB at home, and identify resources to pay for caregiving needs.

*Research should look into what support is needed for the patient, and what support is needed for the caregiver.*

(CG14, wife)

*Depression. It's a life-altering diagnosis. . . It's been very difficult. I put on a face for him because he wants everybody to be happy and pretend that nothing's wrong, but it's hard. It's very hard. . . [Research on how caregivers] are supposed to deal with a diagnosis like Lewy body. I mean, I want to help, but you're in a well of your own to begin with.*

(CG10, wife)

## Research on improving education

Almost all caregivers and half of participants with DLB described a need for improved education surrounding DLB and how research could improve educational efforts. While these opinions were provided in response to questions about research, it was sometimes unclear whether participants were expressing a need for more education clinically or a specific desire for research on this topic. Participants described a need for education in general and also specifically for physicians, medical trainees, therapists, patients, and caregivers/families. The most commonly described needs were education for non-specialist healthcare professionals and for caregivers.

*I think there should be probably some more, uh, I guess they have to have yearly training or whatever, you know. I think that should be something that's sort of at the forefront, because so many doctors don't even know what you're talkin' about when you tell 'em your disease. And, you know, all of them, a lot of 'em.*

(P8)

*Education for regular doctors would be really good.*

(CG19, wife)

*[It] would be helpful for everybody to have a caregiver course, to help take care of the person, but also to learn how to take care of themselves.*

(P25)

*I guess one thing would be to educate the caregiver on the—on the situations that are gonna arise with this disease. . . what to expect and things to address to maybe help the patient.*

(CG16, wife)

## Ranking research priorities

Most of the research topics queried in the interviews or mentioned by interviewees were identified by at least some participants as their top priorities for DLB research (Table 4). Symptoms described as priorities for research included cognition, hallucinations, movement, mobility, sleep, and mood. A few participants mentioned that obtaining the patient and caregiver voices is the most important thing to them.

**Table 4. Top research priorities of individuals with dementia with Lewy bodies and caregivers.**

| Top Research Priorities | Exemplar Quotes |
|---|---|
| Focusing on awareness | Maybe putting some money into getting people more aware of it. (CG14, wife) |
| Determining the cause | Figure out what is goin' on. . . and what Lewy body is, and I'm just not sure about what is goin' on. . .and why it's causing so many changes. (P24) |
| | I would give the whole thousand dollars right into the protein. Because I think everything else that we talked about can be dealt with. . . if it's the protein in the brain that is making lack of or too much, that is making these poor people get this disease, that's what should be addressed. Because everything else would be taken care of. (CG4, wife) |
| Improving diagnosis | I don't think $1,000.00 would cover any of it. Oh, God, I don't know. I would say early—finding the symptoms earlier and hopefully they could see what leads up to it. Yeah, and this way here they could find out, you know, what is the symptom on? Is your-like, is your hand shakin'? Has your eyes changed? Has your skin tone changed?. . .. You know, there could be some kind of a telltale sign. (P4) |
| | I think I'd spend 50 percent of it on the diagnosis and understanding her conditions, which are done by testing, and your laboratories. . . (CG1, husband) |
| | The imaging research I would say is very, very important. I would like to see that progress. . . (P28) |
| Investigating what to expect and disease stages | What is expected with each passing stage, as it appears? (CG12, wife) |
| DLB symptoms needing additional research | To me the most important is his cognition. (CG23, wife) |
| | On movement and—no, I think if you spread [the research money] too thin, that's not good, so. . .I mean, really. (P22) |
| | They need to do research on the physical. (CG19, wife) |
| | *How to practically handle DLB symptoms in daily life*: |
| | And the rest [of the research money] on coping strategies and how to lessen the symptoms as much as possible and increase the quality of life. (CG22, husband) |
| Need for therapies to prevent, cure, or slow DLB | The most important thing would be tryin' to make sure there is a cure for this horrible disease for anyone it affects, not just my loved one, but anybody that is affected by it. (CG7) |
| Research on daily function, daily life, and quality of life | I would divide it number one on quality of life and specifically into the hallucinations and the sleeping. (P8) |
| | I would spend 25% on supportive improvement of lifestyle. (P18) |
| | Quality of life. A hundred percent. . . Having an overarching focus on quality of life, your symptoms answer themselves. For us, it's all quality of life. (CG5, husband) |
| | 50 percent towards quality of life. . .I think it all really boils down to the daily ins and outs. (CG12, wife) |
| Research on caregiving | I guess probably, how, how we can help the caregivers be more successful, or have better lives—as opposed to brain changes. (P20) |
| | Caregiver respite—all of it [the research money]. . .. If the caregiver's not well, nobody's well. . . Ways to incorporate respite for the caregiver or the patient. . . I think caregiver patient respite I would spend the whole $1,000 on that. . . (CG8, wife) |
| Research on improving education | If you can't—you ain't got no knowledge or nothin', none of the rest of it would matter to you anyway. (P17) |
| | Education for patients and for the doctors. Nurses, psychologists, all those people. Well, I mean if I just had $1,000.00 that would be the most important thing to me. (P25) |
| | I would say from my point of view, I would say that we really need more education for other—for the people around [patient] (CG1, husband). |
| | I would do at least a quarter of it in information to the patient and the caregiver—just 'cause I think it's so important. (CG20, wife) |

*The most important to me is that, for you reaching out to find out, you know, who are about this disease and from individual people and how they are feeling—you know, rather than just what's in—you know, you're reading in the books and stuff and to see it and hear it and talk about it for yourself, you know, I think it's very important.*

(CG3, wife).

## Discussion

The study guide queried a variety of research categories and individuals with DLB and caregivers identified topics important for research in all of them—focusing on awareness, determining the cause of DLB, improving diagnosis, investigating what to expect and disease stages, DLB symptoms needing additional research, therapies to prevent, cure, or slow the progression of DLB, targeting daily function and quality of life, caregiving, and improving education. Furthermore, when participants were prompted to assign virtual money to research topics and identify their highest research priorities, at least one participant endorsed each of these areas. This highlights that research is needed across the DLB spectrum—basic science, translational, biomarker, natural history, drug development/therapeutic, quality improvement, and outcomes.

Published Lewy body dementia research priorities were developed primarily by experts and focus on disease mechanisms and processes (including animal model development), diagnostic criteria, terminology, risk factors/prodromal stages, longitudinal cohorts, pathological staging, imaging, biomarkers, genetics, new treatment targets, and preventative/disease-modifying and symptomatic treatments [5, 30]. The individuals with DLB and caregivers in this study supported and prioritized research on many of these topics. However, many topics proposed in this study were outside the national research priorities, possibly because the ADRD summit aimed to inform prevention and effective treatment of AD/ADRDs [5]. Individuals with DLB and caregivers described needing symptomatic treatments but also more research on topics pertaining to how families should handle DLB symptoms in daily life, improving daily life in general, targeting quality of life, caregiving research, understanding disease progression, and improving education regarding DLB across populations (public, patients, caregivers, healthcare professionals).

Recommendations form the 2017 National Research Summit on Care, Services, and Supports for Persons with Dementia and Their Caregivers address some of the themes from current study participants—the need for research to understand different dementia trajectories, determinants of behavioral and psychological symptoms, caregiving, decision-making, and support and financial considerations, as well as research investigating pharmacologic and nonpharmacologic treatment strategies and studying complex, multicomponent programs accommodating care needs [31]. The care summit also identified the need to understand what outcomes are important to persons living with dementia and to engage persons living with dementia and caregivers as part of research teams [31]. Nearly half of the final research recommendations expressed ideas contributed by the persons living with dementia summit working group [6]. This emphasizes the need for funders (federal and otherwise) to consider research priorities from both dementia summits (ADRD and caring) when developing funding announcements and selecting projects to support.

In this study, individuals with DLB and caregivers described several topics as needing research which professionals might put in the category of clinical care. For example, the need for more education across populations was a common theme. Many participants mentioned

needs relating to daily life (decision making, independence including driving, socialization), practical caregiving considerations, when and how to obtain and use extra support, and finances. These practical research priorities are consistent with previously published research priorities of individuals with dementia and their caregivers. The top six overall research priorities identified through modified Delphi consensus involving participants from dementia advocacy organizations were (1) cure and treatment, (2) caregiving, (3) education and training, (4) quality of life, (5) complementary therapies, and (6) care settings [25]. Similarly, dementia priority setting through the James Lind Alliance identified 10 research priorities, including maintaining independence, optimal care, pharmacologic and non-pharmacologic care, and caregiver support [24]. A more recent dementia priority setting process in Canada identified 10 research priorities including investigating stigma, supporting emotional well-being, early treatment, health system changes, caregiver support, connection to education and support, necessary dementia-related skills and knowledge for healthcare professionals, dementia-friendly communities, best dementia practices, and non-pharmacologic treatments [27]. Research priority setting for people with PD identified needed research on physical functioning, symptoms, coping, stress, socialization, relationships, support, autonomy, and good care and communication alongside prioritizing research addressing PD causes, diagnosis, subtypes and medication for both motor and non-motor symptoms [22, 23].

Differences in research priority-setting studies likely reflect the populations participating (e.g. country of origin; combination of patients, caregivers, advocates, healthcare professionals), the types of dementia represented, and the focus of the organizers or project leadership. The current study is also distinct from prior formal priority-setting work, as the goal of the current project was to investigate the views of individuals with DLB and caregivers at a single center, and not to use survey or formal consensus-building processes to build a top-ten list. The current results could serve as background for such a project specific to DLB. Regardless of methodology, it is clear from the current study and prior work that individuals with dementia and caregivers of individuals with dementia, including DLB, prioritize research targeting issues relating to living with disease alongside efforts to better understand the cause of DLB and identify a cure.

Indeed, research into the topics discussed by participants is sorely needed. It is estimated that that 1 in 3 cases of DLB may be missed [32] and initial misdiagnosis is common [32, 33]. It takes over a year for half of individuals with DLB to receive a diagnosis [33]. Family members of individuals who died with DLB described both lack of knowledge of what to expect and negative experiences relating to lacking healthcare professional education and knowledge [12, 34, 35]. While α-synuclein deposition has been the presumed pathogenic cause of DLB for years, recent re-analysis questions this assumption [36]. There are no biomarkers for DLB and no treatments for DLB approved by the U.S. Food and Drug Administration. Caregiver burden in DLB is high [17–19] and quality of life is negatively affected by DLB [20, 21].

This study is the first to evaluate the research priorities of individuals living with DLB and caregivers of individuals with DLB. This study can inform survey development to identify research priorities of a larger group of individuals living with DLB, serve as the background for formal research priority-setting in DLB, and help guide research planning. The study recruited individuals with DLB and caregivers from a single United States-based center of excellence, affects generalizability, though the identified topics were consistent with publications enrolling stakeholders with other dementias and PD. It is plausible that a different cohort could have identified different or additional themes. For example, because enrolled individuals

with DLB had to have mild-moderate dementia (such that they could participate), they and the caregivers who enrolled with them as part of dyads may not have considered end of life research questions, which could potentially be a priority for families dealing with advanced DLB. Furthermore, interview questions were open-ended within categories and additional prompting regarding research topics could have resulted in different responses. The study specifically sought the opinions of individuals with DLB to give them an active voice and role, but several of the participants struggled to understand and answer the questions or perseverated on certain answers, limiting the information that could be gained from those interviews. Most participants were of white non-Hispanic backgrounds and had high educational attainment, potentially affecting generalizability. Similarly, most participating caregivers were wives and it is plausible that other caregiving roles (e.g. husbands, children caring for parents) would have different priorities.

Individuals with DLB and caregivers of individuals with DLB identified areas needing more research related to DLB awareness, causation, diagnosis, prognosis and DLB stages, core and supportive symptoms, disease-modifying and symptomatic treatments, issues relating to daily function and quality of life, caregiving, and education across populations. These findings support the research priorities defined in the NIH care summits in addition to the NIH ADRD summits. Further research is needed across all domains of DLB. Research funding should be informed by the priorities of all relevant stakeholders and support research investigating causes, natural history, biomarkers, and treatment in addition to research targeting themes regarding living with this disease (e.g. independence, quality of life, caregiving, and education).

## Supporting information

**S1 Checklist. COREQ checklist.** COREQ 32-item checklist outlining the page where each element of qualitative research is reported.
(DOCX)

**S1 File. Semi-structured interview guide.**
(DOCX)

**S2 File. Study coding.** Codebook with themes, subthemes, and supporting quotes (with respondent identifiers removed to preserve participant confidentiality).
(XLSX)

## Author Contributions

**Conceptualization:** Melissa J. Armstrong, Tabassum Majid, Angela Taylor, Andrea M. Kurasz, Glenn Smith.

**Data curation:** Melissa J. Armstrong, Noheli Gamez, Slande Alliance.

**Formal analysis:** Melissa J. Armstrong, Noheli Gamez, Slande Alliance.

**Funding acquisition:** Melissa J. Armstrong.

**Investigation:** Melissa J. Armstrong.

**Methodology:** Melissa J. Armstrong, Tabassum Majid, Angela Taylor, Andrea M. Kurasz, Bhavana Patel, Glenn Smith.

**Project administration:** Melissa J. Armstrong, Noheli Gamez, Slande Alliance.

**Resources:** Melissa J. Armstrong.

**Supervision:** Melissa J. Armstrong.

**Validation:** Tabassum Majid, Angela Taylor, Andrea M. Kurasz, Bhavana Patel, Glenn Smith.

**Writing – original draft:** Melissa J. Armstrong.

**Writing – review & editing:** Melissa J. Armstrong, Noheli Gamez, Slande Alliance, Tabassum Majid, Angela Taylor, Andrea M. Kurasz, Bhavana Patel, Glenn Smith.

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
