## [Decision Letter · Decision Letter 0]

22 Jul 2020

PONE-D-20-18287

Research Priorities of Caregivers and Individuals with Dementia with Lewy Bodies: An Interview Study

PLOS ONE

Dear Dr. Armstrong,

Thank you for submitting your manuscript to PLOS ONE. After careful consideration by 2 Reviewers and an Academic Editor, there is a somewhat discrepant view of the submission. Accordingly, all of the critiques of Reviewer #2 must be addressed in detail in a revision to determine publication status. If you are prepared to undertake the work required, I would be pleased to reconsider my decision, but revision of the original submission without directly addressing the critiques of Reviewer #2 does not guarantee acceptance for publication in PLOS ONE. A revised submission will be sent out for re-review. The authors are urged to have the manuscript given a hard copyedit for syntax and grammar.

**Comments to the Author**

1. Is the manuscript technically sound, and do the data support the conclusions?

Reviewer #1: Yes

Reviewer #2: Partly

2. Has the statistical analysis been performed appropriately and rigorously? 

Reviewer #1: Yes

Reviewer #2: N/A

3. Have the authors made all data underlying the findings in their manuscript fully available?

Reviewer #1: Yes

Reviewer #2: No

4. Is the manuscript presented in an intelligible fashion and written in standard English?

Reviewer #1: Yes

Reviewer #2: Yes

5. Review Comments to the Author

Reviewer #1: This is an important and well written study, and gives new perspectives about carer and persons with DLB. Hopefully there will be follow up studies, that also can look into if there are any differences as the DLB progress.

Reviewer #2: The study evaluates the research priorities of DLB patients and their caregivers.

It is an original topic in line with the important topic of consumers’ empowerment which should include decisions in research planning, as well as care.

As far as the “Review Questions” are concerned:

1.One of the aim of the researchers was to address possible significant differences between DLB and other dementias or parkinsonisms, since prior publications reporting on research priorities in PD or “dementia” could not have captured the priorities specific for DLB. However,I was not able to understand what specific issues, different form dementia/parkinsonism in general, emerged in this small cohort.I suggest to discuss in depth whether you found differences in research priorities/claims which are specific to DLB.

A great limit of the study is that it is a single-center based interview study; this fact deeply affects generalizability.It should be stressed that the study has the characteristics of a pilot/feasibility study to assess methodology and important parameters and issues that are needed to design a study with a larger sample population with more possibilities to generalize the results, in particular the saturation of all themes of interests.

As a clinician, I have some difficulties in judging the technical correctness of the qualitative descriptive approach used. I suggest to describe more in depth the methodology quoting original references and not referring to similar studies (ref. 16 and 17). Describing that Microsoft Word® and Excel 2016® were used to organize data and themes is pleonastic.

The following statements seem to be contradictory: if “this study is the first to evaluate the research priorities of individuals living with DLB” (I can’t confirm this...I did not review literature systematically) you cannot write that “the identified topics were consistent with other Publications”.

2.Since a qualitative descriptive approach was used, quantitative statistical analysis was not carried out. See point 1 for the need of more details on qualitative approach and data analysis.

3.Since the qualitative nature of the study, it is important to have some examples from the interviews but not all data underlying the findings described.

Other comments:

Row 79: I suggest “may” fail ...

Row 88: I suggest to use the term Lewy Body Disease (LBD according to professor Kosaka)

Row 92 I suggest to use the term “dream enactment behavior”

Rows 437-8 “Caregiver burden in DLB is high [17-19] and quality of life is worse in DLB compared to AD dementia [33, 34].”I suggest to move this phrase with adequate correction in Introduction where this concept has been already given.

You wrote that individuals with DLB and their caregivers offered many topics pertaining to make medical decisions and strategize long-term planning. In your interview came out the need of tools to enhance Advance Care Planning (ACP)? Did the topic of managing end-of-life issues emerge?

I would like to know whether the topics of ACP and Palliative Care (PC) emerged in the interviews, and how these topics were classified: as “quality of life” or “what to expect and disease stages”, “decision making”, “support to caregivers” or “Long term plans” (fig 1). Why not under a specific topic “ACP”? Please, discuss about it. If these topics did not emerge, discuss why. I think it will be of interest to know if there are some cultural issues to deal with ACP and PC in disease like DLB in your geographical/social context. I was impressed by the fact that two caregivers “didn’t see value in earlier diagnosis without disease-modifying treatments”. This fact seems to open the question that early diagnosis may not be considered important for defining ACP and, more in general,advance provisions for life (and death) preferences before the patient progressed to a dementia level which is not compatible with a decision/preference making.

6. PLOS authors have the option to publish the peer review history of their article (what does this mean?). If published, this will include your full peer review and any attached files.

**Do you want your identity to be public for this peer review?** For information about this choice, including consent withdrawal, please see our Privacy Policy.

Reviewer #1: **Yes: **Ellen Svendsboe

Reviewer #2: **Yes: **Eugenio Pucci

We look forward to receiving your revised manuscript.

Kind regards,

Stephen D. Ginsberg, Ph.D.

Section Editor

PLOS ONE

"I have read the journal's policy and the authors of this manuscript have the following competing interests: MJA receives research support from ARHQ (K08HS24159), a 1Florida ADRC pilot grant (AG047266), the Florida Department of Health Ed & Ethel Moore research program, and as the local PI of a Lewy Body Dementia Association Research Center of Excellence. AT is employed by the Lewy Body Dementia Association. BP receives research support from an American Academy of Neurology Clinical Research Training Scholarship in Lewy Body Dementia. GS receives research support from the 1Florida ADRC (AG047266) and the Florida Department of Health Ed & Ethel Moore research program."

---

## [Author Response · Author response to Decision Letter 0]

17 Aug 2020

[This was uploaded as a separate file with formatting but we are also cutting and pasting the response below.]

Reviewer #1:

Reviewer: This is an important and well written study, and gives new perspectives about carer and persons with DLB. Hopefully there will be follow up studies, that also can look into if there are any differences as the DLB progress.

Response: Thank you. (Nothing to do.)

Reviewer #2:

Reviewer: The study evaluates the research priorities of DLB patients and their caregivers.

It is an original topic in line with the important topic of consumers’ empowerment which should include decisions in research planning, as well as care.

Response: N/A

Reviewer: As far as the “Review Questions” are concerned:

1.One of the aim of the researchers was to address possible significant differences between DLB and other dementias or parkinsonisms, since prior publications reporting on research priorities in PD or “dementia” could not have captured the priorities specific for DLB. However,I was not able to understand what specific issues, different form dementia/parkinsonism in general, emerged in this small cohort.I suggest to discuss in depth whether you found differences in research priorities/claims which are specific to DLB.

Response: To clarify, the aim of the project was “to identify the research priorities of individuals with DLB and caregivers…” (pages 4-5). It was not an aim to address possible significant differences between DLB and other dementias or parkinsonisms (this would need a separate design, such as a study enrolling all 3 populations). The reference to the other diseases in the introduction is because the authors felt that the priorities in DLB could be different from other dementias and Parkinson disease because of meaningful differences in disease experiences and trajectories, and this provided the rationale for the current project. While addressing differences between diseases was not an aim, we agree that this is important for the discussion. Some of this information was already present, but we have edited the paragraph to reflect both of the referenced PD publications and not just the dementia publications: “These practical research priorities are consistent with previously published research priorities of individuals with dementia and their caregivers. The top six overall research priorities identified through modified Delphi consensus involving participants from dementia advocacy organizations were (1) cure and treatment, (2) caregiving, (3) education and training, (4) quality of life, (5) complementary therapies, and (6) care settings [23]. Similarly, dementia priority setting through the James Lind Alliance identified 10 research priorities, including maintaining independence, optimal care, pharmacologic and non-pharmacologic care, and caregiver support [22]. A more recent dementia priority setting process in Canada identified 10 research priorities including investigating stigma, supporting emotional well-being, early treatment, health system changes, caregiver support, connection to education and support, necessary dementia-related skills and knowledge for healthcare professionals, dementia-friendly communities, best dementia practices, and non-pharmacologic treatments [25]. Research priority setting for people with PD dementia identified needed research on physical functioning, symptoms, coping, stress, socialization, relationships, support, autonomy, and good care and communication alongside prioritizing research addressing PD causes, diagnosis, subtypes and medication for both motor and non-motor symptoms [22, 23].” (page 22, lines 418-419, tracked changes version)

Reviewer: A great limit of the study is that it is a single-center based interview study; this fact deeply affects generalizability.It should be stressed that the study has the characteristics of a pilot/feasibility study to assess methodology and important parameters and issues that are needed to design a study with a larger sample population with more possibilities to generalize the results, in particular the saturation of all themes of interests.

Response: We have edited the sentence in limitations to be clear that the subject group affects generalizability: “The study recruited individuals with DLB and caregivers from a single United States-based center of excellence, which affects generalizability, though the identified topics were consistent with publications enrolling stakeholders with other dementias and PD. It is plausible that a different cohort could have identified additional themes.” (Page 23, lines 450-451, tracked changes version). We agree that one hope of these results is to inform planning of a larger study: “This study can inform survey development to identify research priorities of a larger group of individuals living with DLB, serve as the background for formal research priority-setting in DLB, and help guide research planning.” (Page 23, lines 445-447, tracked changes version). This study was not designed to assess methodology, however; it used an established qualitative methodology technique (now better reflected in the correct references, see below).

Reviewer: As a clinician, I have some difficulties in judging the technical correctness of the qualitative descriptive approach used. I suggest to describe more in depth the methodology quoting original references and not referring to similar studies (ref. 16 and 17). Describing that Microsoft Word® and Excel 2016® were used to organize data and themes is pleonastic.

Response: We apologize; there was an error in our references and this was indeed very confusing. The references 16 and 17 in the submitted manuscript were incorrect; it appears that they were converted to plain text too early so that the original references were lost. We have added the appropriate references: Colorafi 2016 describes qualitative descriptive approaches and Tong 2007 is the reference for reporting qualitative studies. We have maintained the reference to the use of Microsoft Word and Excel because item #27 of the COREQ checklist for qualitative research reporting (S1 checklist) is the software used (i.e., reporting the software use is parting of the reporting checklist for qualitative studies). While the use of Word and Excel is somewhat mundane, we are required to clarify that we did not use some of the purchasable qualitative research software available such as NVIVO. (Methods section, “Study design,” page 5, tracked changes version.) We select qualitative software by project and this one was served well by Word and Excel and did not require advanced qualitative software packages.

Reviewer: The following statements seem to be contradictory: if “this study is the first to evaluate the research priorities of individuals living with DLB” (I can’t confirm this...I did not review literature systematically) you cannot write that “the identified topics were consistent with other Publications”.

Response: We have clarified the sentence to read “…, though the identified topics were consistent with publications enrolling stakeholders with other dementias and PD” (page 23, line 450, tracked changes version).

Reviewer: 2.Since a qualitative descriptive approach was used, quantitative statistical analysis was not carried out. See point 1 for the need of more details on qualitative approach and data analysis.

Response: We have now added the correct reference for the qualitative descriptive response (“study design,” first paragraph of methods, page 5 of tracked changes version). The correct reference was also added to the “data collection and analysis” section (line 151, tracked changes version). Details of how the qualitative analysis was performed is also captured in this section (page 7, lines 150-160, tracked changes version): “Investigators used tables in Microsoft Word® and Excel 2016® to organize data and a qualitative descriptive approach to identify and organize themes [26]. Broad topics/categories were defined by interview questions, but themes were identified from interview transcripts. The PI and two research assistants independently analyzed interview transcripts to create a codebook and then reached consensus regarding emerging themes (open coding). The research assistants analyzed remaining transcripts using a constant comparative technique, revising themes and subthemes with the PI if needed (axial coding) (S3 File). Coders assessed saturation during analysis. Co-investigators gave feedback after the initial coding. Participants were numbered in the analysis such that participants who enrolled in the study as a dyad shared participant numbers (with “P” indicating patient participants and “CG” indicating caregiver participants).”

Reviewer: 3.Since the qualitative nature of the study, it is important to have some examples from the interviews but not all data underlying the findings described.

Response: There are different approaches to qualitative analyses. Some qualitative analyses simply state the synthesized ideas without many quotes. A qualitative descriptive approach, however, presents quotes that provide examples for the identified themes/subthemes. The examples included do not reflect all the underlying data, but serve as illustrative quotes. We tried to include, where possible, one quote from a patient and one quote from a caregiver to demonstrate these themes across populations (where appropriate). In response to this comment and the editor’s comment, we have now included a de-identified supplemental Excel file (S3 File) showing the underlying data. 

Reviewer: Other comments: Row 79: I suggest “may” fail ...

Response: We have not made this change because by definition, consultation strategies do not give patient representatives an active voice in the process. Consultation strategies ask for opinions but do not actively include a representative as a co-participant. If a person has an active voice in the process, that is by definition a “participation” strategy.

Reviewer: Row 88: I suggest to use the term Lewy Body Disease (LBD according to professor Kosaka)

Response: We have revised the sentences to read, “DLB is a subset of Lewy body dementia, the 2nd most common neurodegenerative dementia in the United States [8]. DLB is a dementia with clinical and pathological overlap with Parkinson disease (PD) and both fall in the pathological category of Lewy body diseases.” (Page 4, lines 88-90, tracked changes version)

Reviewer: Row 92 I suggest to use the term “dream enactment behavior”

Response: Changed to “dream enactment behavior” as suggested (page 4, line 93, tracked changes version)

Reviewer: Rows 437-8 “Caregiver burden in DLB is high [17-19] and quality of life is worse in DLB compared to AD dementia [33, 34].”I suggest to move this phrase with adequate correction in Introduction where this concept has been already given.

Response: We have added the sentence “Quality of life is also worse in DLB compared to AD dementia [20, 21]” to the introduction on page 4 as suggested (line 97, tracked changes version). We have no fully removed these concepts from the discussion, because this paragraph is providing context for why research into the topics discussed by participants is needed. We have reworded the paragraph (page 23, lines 433-442, tracked changes version) to read: “Indeed, research into the topics discussed by participants is sorely needed. It is estimated that that 1 in 3 cases of DLB may be missed [32] and initial misdiagnosis is common [32, 33]. It takes over a year for half of individuals with DLB to receive a diagnosis [33]. Family members of individuals who died with DLB described both lack of knowledge of what to expect and negative experiences relating to lacking healthcare professional education and knowledge [12, 34, 35]. While α-synuclein deposition has been the presumed pathogenic cause of DLB for years, recent re-analysis questions this assumption [36]. There are no biomarkers for DLB and no treatments for DLB approved by the U.S. Food and Drug Administration. Caregiver burden in DLB is high [17-19] and quality of life is negatively affected by DLB [20, 21].”

Reviewer: You wrote that individuals with DLB and their caregivers offered many topics pertaining to make medical decisions and strategize long-term planning. In your interview came out the need of tools to enhance Advance Care Planning (ACP)? Did the topic of managing end-of-life issues emerge?

Response: Advance care planning came up in the portion of the interviews about clinical care (the first half of the interviews), but not in the portion of the interviews about research priorities, which is the topic of the current analysis and manuscript. Sadly to us, since several of the researchers on this paper research the end of life in dementia with Lewy bodies, this was not a major theme in the voiced research priorities.

Reviewer: I would like to know whether the topics of ACP and Palliative Care (PC) emerged in the interviews, and how these topics were classified: as “quality of life” or “what to expect and disease stages”, “decision making”, “support to caregivers” or “Long term plans” (fig 1). Why not under a specific topic “ACP”? Please, discuss about it. If these topics did not emerge, discuss why. I think it will be of interest to know if there are some cultural issues to deal with ACP and PC in disease like DLB in your geographical/social context. I was impressed by the fact that two caregivers “didn’t see value in earlier diagnosis without disease-modifying treatments”. This fact seems to open the question that early diagnosis may not be considered important for defining ACP and, more in general,advance provisions for life (and death) preferences before the patient progressed to a dementia level which is not compatible with a decision/preference making.

Response: As noted above, ACP and palliative care were not mentioned during the research portions of the interviews. While we agree that these topics overlap with some of the other themes mentioned, ACP and palliative care as specific research topics were not identified themes. We have added a discussion of this to the limitations section (pages 23-24, tracked changes version): “It is plausible that a different cohort could have identified different or additional themes. For example, because enrolled individuals with DLB had to have mild-moderate dementia (such that they could participate), they and the caregivers who enrolled with them as part of dyads may not have considered end of life research questions, which could potentially be a priority for families dealing with advanced DLB. Furthermore, interview questions were open-ended within categories and additional prompting regarding research topics could have resulted in different responses.”

Journal Requirements/Other Comments:

Response: We used PACE and have revised our image accordingly.

Response: We re-reviewed the style requirements and made a couple small adjustments.

"I have read the journal's policy and the authors of this manuscript have the following competing interests: MJA receives research support from ARHQ (K08HS24159), a 1Florida ADRC pilot grant (AG047266), the Florida Department of Health Ed & Ethel Moore research program, and as the local PI of a Lewy Body Dementia Association Research Center of Excellence. AT is employed by the Lewy Body Dementia Association. BP receives research support from an American Academy of Neurology Clinical Research Training Scholarship in Lewy Body Dementia. GS receives research support from the 1Florida ADRC (AG047266) and the Florida Department of Health Ed & Ethel Moore research program."

Response: We re-reviewed all the authors’ disclosures (one small adjustment not relevant to the paper) and added the requested statement. This is all in the cover letter.

Response: We did not originally upload the full analysis because of concerns that the participants could be identifiable. To meet this requirement, we have removed the participant attribution from each quote in the coding. We have now added the S3 file with the codebook and the underlying quotes (but without the attribution for each quote, which we maintain).

---

## [Decision Letter · Decision Letter 1]

3 Sep 2020

Research Priorities of Caregivers and Individuals with Dementia with Lewy Bodies: An Interview Study

PONE-D-20-18287R1

Dear Dr. Armstrong,

We’re pleased to inform you that your manuscript has been judged scientifically suitable for publication and will be formally accepted for publication once it meets all outstanding technical requirements.

Kind regards,

Stephen D. Ginsberg, Ph.D.

Section Editor

PLOS ONE

**Comments to the Author**

1. If the authors have adequately addressed your comments raised in a previous round of review and you feel that this manuscript is now acceptable for publication, you may indicate that here to bypass the “Comments to the Author” section, enter your conflict of interest statement in the “Confidential to Editor” section, and submit your "Accept" recommendation.

Reviewer #1: All comments have been addressed

Reviewer #2: All comments have been addressed

2. Is the manuscript technically sound, and do the data support the conclusions?

Reviewer #1: Yes

Reviewer #2: Yes

3. Has the statistical analysis been performed appropriately and rigorously? 

Reviewer #1: Yes

Reviewer #2: N/A

4. Have the authors made all data underlying the findings in their manuscript fully available?

Reviewer #1: Yes

Reviewer #2: Yes

5. Is the manuscript presented in an intelligible fashion and written in standard English?

Reviewer #1: Yes

Reviewer #2: Yes

6. Review Comments to the Author

Reviewer #1: Looking forward to read more about this topic and hoping the authors have further plans for following up studies!

Reviewer #2: I'm grateful to the Authors for addressing the topics of ACP and PC. The fact that ACP and PC were not mentioned during the interviews as specific research topics deserves further attention and discussion beyond this paper's aims. I will be glad to further discuss and disseminate this result, even in a further publication aimed at trying to gain an in-depth understanding of why ACP and PC did not emerge as specific topics.

7. PLOS authors have the option to publish the peer review history of their article (what does this mean?). If published, this will include your full peer review and any attached files.

Reviewer #1: **Yes: **Ellen J. Svendsboe

Reviewer #2: **Yes: **Eugenio Pucci

---

## [Editor Report · Acceptance letter]

15 Sep 2020

PONE-D-20-18287R1

Research Priorities of Caregivers and Individuals with Dementia with Lewy Bodies: An Interview Study

Dear Dr. Armstrong:

I'm pleased to inform you that your manuscript has been deemed suitable for publication in PLOS ONE. Congratulations! Your manuscript is now with our production department.

Kind regards,

on behalf of

Dr. Stephen D. Ginsberg 

Section Editor

PLOS ONE